# Abrupt events and population synchrony in the dynamics of Bovine Tuberculosis

Aristides Moustakas[1], Matthew R. Evans[2], Ioannis N. Daliakopoulos[3,4] & Yannis Markonis [5]

Disease control strategies can have both intended and unintended effects on the dynamics of infectious diseases. Routine testing for the harmful pathogen Bovine Tuberculosis (bTB) was suspended briefly during the foot and mouth disease epidemic of 2001 in Great Britain. Here we utilize bTB incidence data and mathematical models to demonstrate how a lapse in management can alter epidemiological parameters, including the rate of new infections and duration of infection cycles. Testing interruption shifted the dynamics from annual to 4-year cycles, and created long-lasting shifts in the spatial synchrony of new infections among regions of Great Britain. After annual testing was introduced in some GB regions, new infections have become more de-synchronised, a result also confirmed by a stochastic model. These results demonstrate that abrupt events can synchronise disease dynamics and that changes in the epidemiological parameters can lead to chaotic patterns, which are hard to be quantified, predicted, and controlled.

[1] Institute for Applied Data Analytics, Universiti Brunei Darussalam, Jalan Tungku Link, Gadong, BE 1410, Brunei Darussalam. [2] School of Biological Sciences, Kadoorie Biological Sciences Building, The University of Hong Kong, Pok Fu Lam Road, Hong Kong, SAR, China. [3] School on Environmental Engineering, Technical University of Crete, Polytechnioupolis, Chania 73100, Greece. [4] Department of Agriculture, Technological Educational Institute of Crete, Estavromenos, Heraklion 71004, Greece. [5] Faculty of Environmental Sciences, Czech University of Life Sciences Prague, Kamýcká 129, Praha-Suchdol 165 00, Czech Republic. Correspondence and requests for materials should be addressed to A.M. (email: arismoustakas@gmail.com)

Patterns of infectious diseases across spatial and temporal scales are fundamental for understanding their dynamics and for designing eradication strategies[1]. Bovine Tuberculosis (bTB) is a transmissible bacterial infection that can spread rapidly within livestock due to chronic shedding and highly variable incubation period posing a serious threat to the agricultural industry and animal wildlife. Historically, bTB was brought to the edge of elimination within cattle in Great Britain (GB) in the late 1970s[2]. Since then it has been steadily increasing in both its prevalence in the cattle herd and its spread around the country[3,4]. This current severe epidemic has devastating effects on the agricultural industry with economic, societal, and ecosystem consequences.

Time-ordered data of natural processes (i.e., data regarding the evolution over time of a phenomenon, sampled in different, sometimes uneven, time steps) are commonly found to be correlated, and thus not independent. While this provides a challenge in the analysis, as issues such as temporal autocorrelation[5] and stationarity[6] need to be accounted for, it is this non-independency that makes the data interesting[7] and may allow potential effects of treatments, trends over time, or abrupt changes to be detected. A subset of treatments is abrupt events; phenomena that rarely do occur, last a relatively short time, and are associated with extreme values—can synchronise patterns, exhibit long-lasting effects[8], possibly resulting in tipping points[9]. The impact of such abrupt events on ecosystem dynamics can be investigated through the analysis of space-time patterns[10].

In GB, during the foot and mouth disease (FMD) epidemic, cattle testing against bTB was interrupted for a relatively short time period: in January 2001 cattle testing, and consequently the management of infections, was interrupted for 10 months due to the outbreak of FMD. From November 2001 and for the following 12 months, cattle testing was focused on high-risk areas and the regular cattle testing that had been taking place nationally was suspended. From November 2002, testing was implemented as pre-FMD outbreak. Cases of bTB are recorded and these have generated a unique data set, allowing the temporal and spatial prevalence of the disease within GB to be followed[11]. Here we analyse a monthly data set on New Herd Incidents (NHI; herds that were detected to contain at least one infected individual having previously been bTB-free) normalised by the number of Total Tests on Herds (TTH) from January 1996 to August 2016[11].

Using this data set, we sought to quantify patterns of the disease spread over space and time. In addition, we examined the impact of epidemiological changes due to the period during which disease controls were suspended indicated by the interruption to cattle testing, an abrupt short-lasting event, on disease prevalence, temporal cycles of infections, and spatial population synchrony. Moreover, we quantified the dissemination response to the external factor of testing frequency on the disease dynamics and control. Here we show that the changes in epidemiological parameters during the short-lasting unmanaged time while testing (and follow-up actions) was suspended, can increase new infections markedly, can have long-lasting effects, and generate longer-term temporal infection cycles. Infection cycles shifted from annual to 4-year after testing interruption. Spatial synchrony of new infections between different GB regions after the interruption of cattle testing increased. These effects persisted for over 15 years after the abrupt testing interruption. After annual testing was introduced in some GB regions, new infections have become more de-synchronised, a result also confirmed by a stochastic model. This study shows that amendments in the epidemiological parameters lead to chaotic patterns and that abrupt events synchronise disease dynamics.

## Results

**Raw data and probability distributions.** NHI increased in GB from January 1996 to December 2000, followed by a steep increase between January 2001 to November 2002. From December 2002 to September 2015, NHI increased at a lower rate than during the period in which testing was interrupted, but the number of NHI were at least an order of magnitude higher than prior to January 2001 (Fig. 1a). TTH increased throughout the time span of the data with the exception of testing interruption period (Fig. 1b). NHI/TTH increased before testing interruption, peaked during the testing interruption, and exhibited no visible monotonic pattern after the testing interruption (Fig. 1c). While NHI and TTH are seasonal (Fig. 1a, b), the NHI/TTH index is not seasonal (Supplementary Fig. 1). The empirical probability distribution of NHI/TTH is similar before and after testing interruption, however, the mean of the distribution is considerably higher after the testing interruption (data from January 2001 to November 2002 are excluded from the analysis); (Fig. 1d). This result in England derived mainly from high-risk areas, where the difference between the means of the NHI/TTH distribution before and after testing interruption is considerably larger (Fig. 1e) than in low-risk areas (Fig. 1f). These results show that the mean of the probability density function (PDF) for high-risk areas has substantially shifted towards larger values of NHI/TTH compared to that for low-risk areas, indicating that infection spread is considerably faster after the testing interruption in high-risk areas. A different regime is observed in Scotland, all counties of which have been officially bTB-free since October 2009. There, the testing interruption had only a marginal effect, as shown by substantial overlap in the PDF plots before and after the interruption (see Supplementary Fig. 2).

**Mean field wavelet analysis of NHI/TTH.** Wavelet mean field analysis of NHI/TTH (Fig. 2) suggests the frequent emergence of high-frequency synchronisation events and synchronous features associated with two slow, statistically significant fluctuations in the mean of all time-series (Supplementary Note 3 and Supplementary Figs. 3, 4, and 5); statistical significance is assessed at a 95% confidence level across all three significance wavelet tests[12,13]. The testing interruption event manifests as the strongest signal when the whole record is taken into account (Fig. 2a; Supplementary Fig. 5a). Before testing interruption, the annual fluctuation of the infection cycle was dominant (Fig. 2b; Supplementary Fig. 5b), and gradually, after the abrupt event, moved to lower frequencies, dispersed around a 4-year mean (Fig. 2c; Supplementary Fig. 5c). The high-frequency synchronisation can be found in the 2- to 4-month band and its duration ranges from couple of months to half a year. For high-risk areas, the synchronicity in some events reaches 1, i.e., there is complete synchronisation of all the counties (Fig. 2d; see also Supplementary Fig. 5d). The weaker 12-month cycle also persisted, but only for one third of the low-risk areas, where both the high and the low-frequency co-variability is negligible (Fig. 2e and Supplementary Fig. 5e). Since the NHI/TTH is not biased in terms of testing seasonality, this annual cycle could be linked with other factors that affect the bTB seasonality such as the annual climatic cycle. Finally, it is noteworthy that after 2013 the high-frequency synchronisation effects decrease sharply to low levels for both high- and low-risk areas. Analysis using the wavelet energy of the mean time series yielded similar to identical results to the ones reported here (Supplementary Fig. 5).

**Spatial synchrony and network analysis.** Analysis regarding the geography of spatial synchrony[14] included the number of nodes above threshold, indicating the number of counties with high

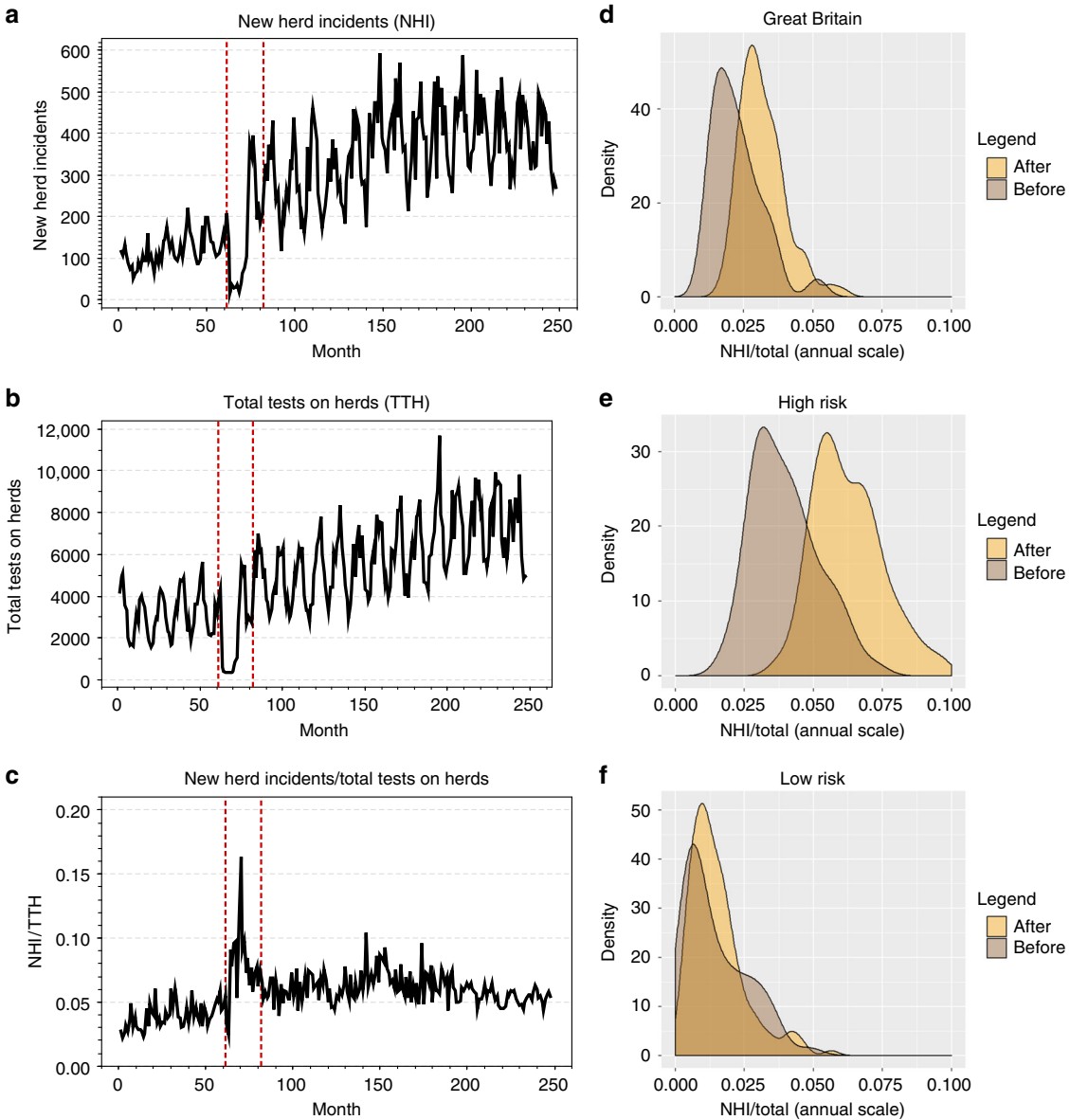

**Fig. 1** Raw data and probability distributions. **a** Time series data of New Herd Incidents (NHI) of Bovine Tuberculosis in cattle in Great Britain from January 1996 to August 2016. The data are recorded every month. Testing interruption period is indicated with dotted vertical red lines. **b** Time series data of Total Tests on Herds (TTH) in GB during the same period. **c** Time series data of NHI/ TTH in GB during the same period. **d** Empirical probability densities for NHI/TTH before and after the interruption in testing in GB. **e** Empirical probability densities for NHI/TTH before and after the interruption in testing in high-risk areas. **f** Empirical probability densities for NHI/TTH before and after the interruption in testing in low-risk areas

NHI/TTH values, and the number of connections between nodes, indicating the synchrony amongst them. NHI/TTH were more spatially synchronised during the testing interruption period than before or immediately after (Fig. 3a–c, Supplementary Figs. 6a–c, and Supplementary Note 4) as indicated by the covariance matrices of differences in NHI/TTH per GB counties in GB after removing it within county annual variation and network analysis. During January 1996–March 2000, there were only 14 counties in GB with both NHI/TTH > 0.3, and at least 6 months with at least one NHI case (thereby counties above threshold—Supplement S3); (Fig. 3a; Supplementary Fig. 6a) and the mean number of connections per county was 2.1. During April 2000–April 2004, a snapshot that contains testing interruption, there were 25 counties above threshold (Fig. 3b; Supplementary Fig. 6b) and the mean number of connections per county was 6.8. During May

2004–June 2008, the period after interruption, there were 32 counties above the threshold (Fig. 3c; Supplementary Fig. 6c) and the mean number of connections per county was 4.9. During July 2008–July 2012, the number of counties above the threshold was 39 (Fig. 3d; Supplementary Fig. 6d) and the mean number of connections per county was 5.9. During the last available period, August 2012–August 2016, the number of counties was 40 (Fig. 3e and Supplementary Fig. 6e) but the mean number of connections per county dropped to 3.1. In England, the number of high-risk counties above threshold has drastically reduced after the introduction of annual testing in high-risk areas in 2013 (Fig. 3f). Synchrony was minimal during the period prior to testing interruption (Fig. 3a–f; Supplementary Fig. 6a), peaked during testing interruption period (Figs. 3b; 4f; Supplementary Fig. 6b) and was again weakened during the last period after

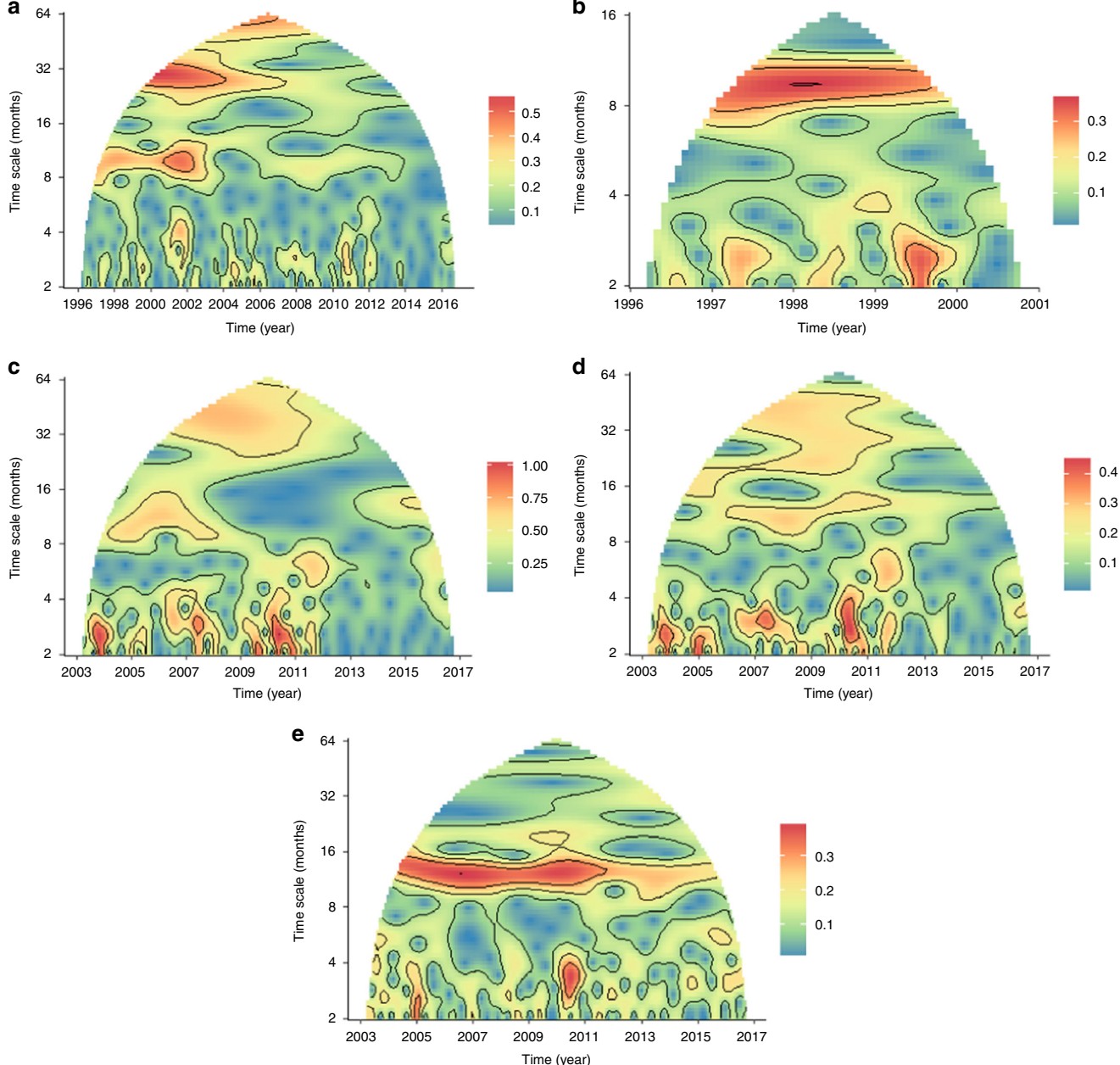

**Fig. 2** Mean Field Wavelet analysis of NHI/TTH time series from January 1996 to August 2016. The vertical axis indicates periods (cycles) while the horizontal time. The brown contours correspond to four equal classes as derived from the legend **a** GB for the whole period, **b** GB before interruption, **c** GB after interruption, **d** High-risk areas after interruption, **e** Low-risk areas after interruption. For a wavelet energy of the mean time series analysis, see Supplementary Fig. 5

implementing annual testing in high risk and edge areas during year 2013 (Figs. 3e; 4f; Supplementary Fig. 6e).

**Comparing data and IBM outputs**. Results from a calibrated individual-based model[15] (IBM), modelling the effects of testing interruption and testing frequency on NHI/TTH, indicated that the IBM was able to reproduce the intervention[16] of annual testing introduction in high risk and edge areas in England in 2013, based on testing frequency alone, all else being equal (Fig. 4a). As indicated by the third panel in Fig. 4a, the IBM slightly over-estimates the impact of the introduction of annual testing but does not differ significantly from the data in terms of the impact of switching from 4-year

to annual testing (Fig. 4a and Supplementary Table 1). Had the intervention of annual testing introduction in 2013 not taken place[17] (all else been equal but testing frequency kept at every 4 years throughout the simulation) NHI/TTH would have been around 15% higher within 3 years (Fig. 4b and Supplementary Table 2).

## Discussion

The earliest recorded NHI/TTH incidents in the data set (Jan 1996) were both few and not synchronised. In terms of time, the main spread of the disease as well as synchrony occurred during testing interruption period. Spatially, NHI/TTH spread from high-risk areas of the South-West to low risk via edge areas

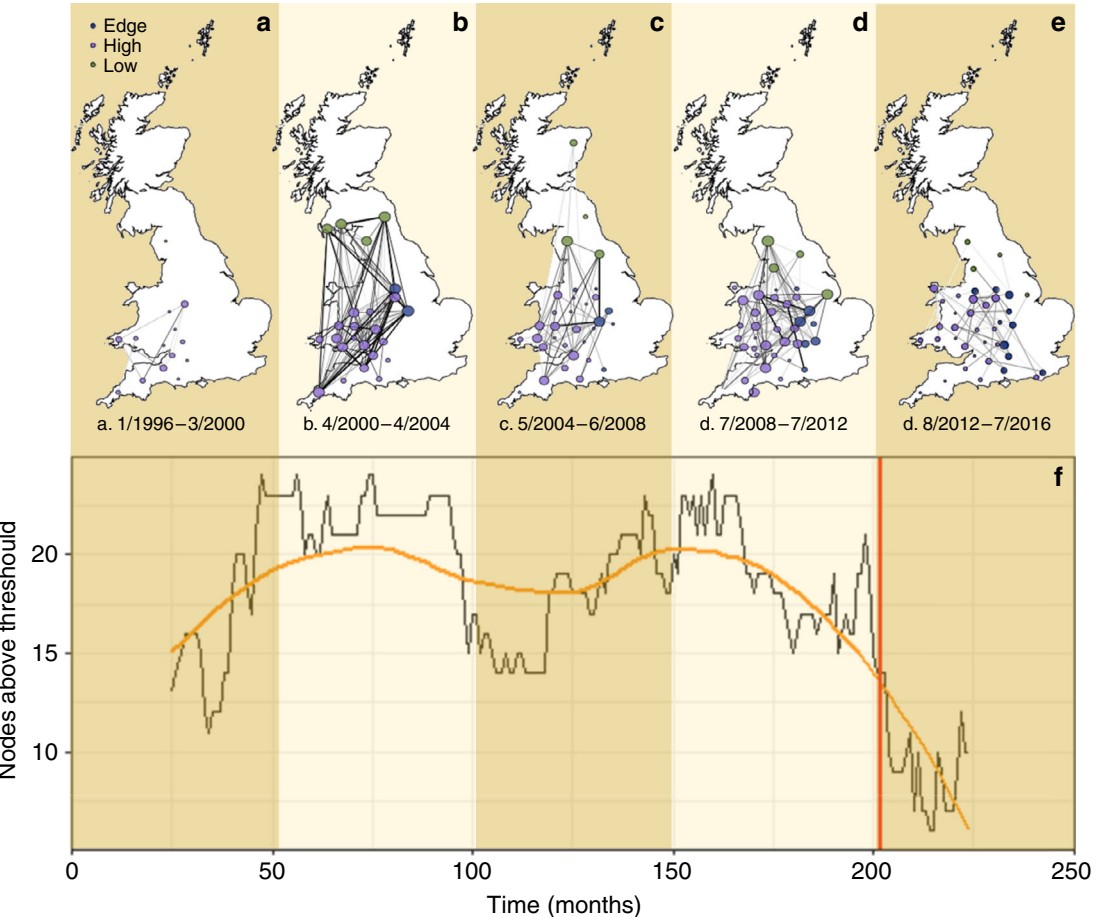

**Fig. 3** Spatial synchrony and Network Analysis. Plotted results of correlation matrixes of differences in NHI/TTH per GB counties after the within the county annual variation has been removed. The numbers of nodes indicate the number of counties with high NHI/TTH values (counties above threshold), while the number of connections between nodes, the synchrony amongst them. The thickness of the connecting line between two nodes is proportional to the correlation value between those counties in the corresponding correlation matrix. Only positive correlations are plotted. Low-risk counties are plotted with green, edge with blue, and high risk with purple colour. The data are temporally partitioned into five snapshots as following: **a** 1/1996-3/2000, **b** 4/2000-4/2004, **c** 5/2004-6/2008, **d** 7/2008-7/2012, and **e** 8/2012-8/2016. **f** Number of above threshold high-risk counties in England throughout the data set. The vertical red line indicates the introduction of annual testing in January 2013. The orange line is derived from a moving average of 48 months centred in mid value loess regression. The foreground data presented on the image was created by the authors using the methodology described in the methods. The background map of GB Contains OS data © Crown copyright and database right (2016). All rights reserved

mainly during testing interruption (see also a moving image of NHI/TTH spread in GB from January 1996 to August 2016 at https://goo.gl/1kKfVq). During the bTB testing interruption period there was an outbreak of FMD and many farms in high FMD incidence areas were left with very few or no cattle at all[18]. Cattle restocking of those farms was conducted via purchasing cattle, sometimes from high-risk bTB areas, which were moved either without compulsory pre- (made compulsory in England in 2006) or post- (made compulsory in England in 2016) cattle movement bTB testing[18]. It should be noted that it is not testing interruption per se that increases infections; the interruption simply provided (a) time availability for unmanaged disease spread within each farm and between neighbouring farms (fine scale), and (b) time availability for unmanaged disease spread via cattle movements to distant farms (coarser scales). Thus, the lack of control during bTB testing interruption changed the epidemiological parameters such as infection rate, dispersal distances, and population densities[18].

Cattle movement is a well-known source of bTB spread[4] but the typical distances of cattle movement in the UK are short[17] and cattle move few times in their life[19]. Most movements between any given pair of farms only occurred once in a study spanning form 1999 to 2009[19]. However, during FMD outbreaks in the same study period, cattle movements increased up to tenfold[19]. Dispersal can substantially increase the scale of population synchrony for weakly regulated populations[20] (as was the case for cattle in GB during testing interruption period) and the spatial scale of time-lagged synchrony increases with species dispersal distance[21]. The time-lagged synchrony is strongest at a species' typical dispersal distance because the rate of exchange between populations is greatest at that distance[21]. This provides a plausible explanation for the shift towards longer distances for bTB synchrony during and immediately after testing interruption.

Wavelet and additional autocorrelation analysis (Supplementary Figs. 3 and 4) indicated that the autocorrelation structure changed dramatically after the interruption. Infection cycles shifted from annual to 4-year after testing interruption. The annual cycle was no longer the dominant component, being replaced by a more persistent behaviour, i.e., the autocorrelation coefficient maintains statistically significant values for larger monthly lags. This behaviour is typical of long-term persistence

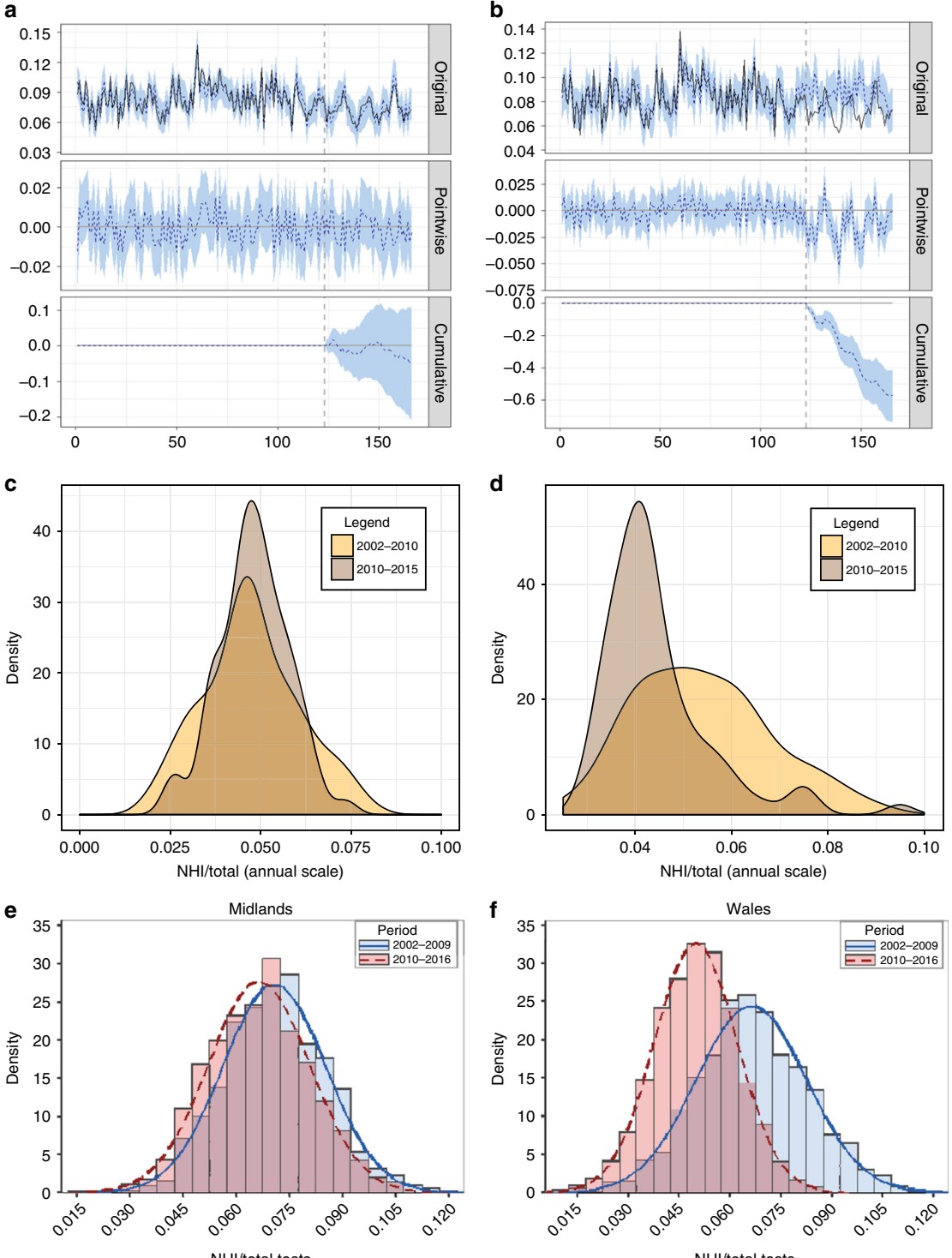

**Fig. 4** Comparing data and IBM outputs. **a** NHI/TTH data and IBM outputs comparisons for English high risk and edge areas after testing interruption with Bayesian inference time series intervention analysis. The 'intervention' point is January 2013 when annual testing was introduced. Analysis for scenario I – testing switched from 4 years to annual in the IBM (test) as it happens in the data (control). The first panel shows the actual test values (black line) against the forecast (counterfactual prediction for the post-intervention period) in a dotted blue line. In all three panels 95% prediction confidence intervals are indicated with shaded light blue colour. The intervention point is indicated with a vertical dashed back line. The second panel shows the difference between observed data and counterfactual predictions. This is the pointwise causal effect, as estimated by Bayesian analysis. The third panel adds up the pointwise contributions from the second panel, resulting in a plot of the cumulative effect of the intervention. **b** Same as **a** for scenario II – testing every 4 years throughout the IBM simulation (test) against the same data (control). **c** Empirical probability densities for NHI/TTH data in English Midlands after the interruption but before the implementation of at least annual testing (2002–2009, yellow colour) and after at least annual testing frequency implementation in Wales (2010–2016, brown colour). **d** Same as **c** for Wales. **e** the empirical probability densities of NHI/TTH from outputs of the parameterised IBM for Midlands. **f** Same as **e** for Wales

processes[22], suggesting enhanced temporal variability. Long-term persistence means there are stochastic fluctuations on all time scales[23], i.e., the autocorrelation function goes to zero very slowly. Long-term persistence is often attributed to be a sign of unforced (internal) variability, i.e., a property of the system regardless of external management[24]. Here both the annual and the 4-year cycles were a property of the disease but the 4-year cycle was dominant in high-risk areas throughout the study period and became more pronounced after testing interruption. Spatial synchrony of new infections between different GB regions after the interruption of cattle testing also increased. Thus, an abrupt event such as the relatively short-lasting unmanaged time provided by the interruption of bTB testing, can increase new bTB infections markedly, generate longer-term infection cycles, and spatiotemporally synchronise new infections. These effects can persist for very long (over 15 years in the current data set), suggesting catastrophic shifts[25] or tipping points[26]. To that end, robust early warning indicators are needed[27].

Results suggest that before the testing interruption, counties in GB except in Scotland (Supplementary Fig. 2) were less spatially synchronised with each other than they were after the interruption. Counties in Wales became more synchronised with counties in English Midland after testing interruption (Supplementary Note 6 and Supplementary Fig. 7). It is important to note that Wales has implemented annual or more frequent cattle testing since January 2010[11,28]. Since then, disease spread was desynchronised between counties in Wales and both the region with which Wales was previously most synchronised (English Midlands), as well as North of England, the region with which Wales was least synchronised (Supplementary Fig. 7). In addition to frequent testing, desynchronising new infections between regions has also changed the distribution of new infections: the PDE of new infections after the introduction of more frequent testing, in Wales and English Midlands (previously the region most synchronised with Wales) differ markedly with new infections being dominated by lower frequencies (implying low NHI/TTH values) in Wales than English Midlands; a result confirmed both by data (Fig. 4c, d and Supplementary Fig. 8) and the parameterised IBM outputs (Fig. 4e, f).

Time series analysis poses great challenges[29] as clustering of extreme events and strong shifts in the record mean could occur more frequently than depicted in the data reconstructions[30]. Imperfect knowledge of time series interactions with factors such as weather and climate features[31], testing sensitivity[32], epidemiological parameters[2] etc., pose limitations to our understanding of the system and the ability to define causation. Nevertheless, in view of evidence such as the following: (i) the spatial de-synchronisation after the introduction of annual testing in high risk and edge English areas after 2013, (ii) the striking contrast between GB and Scotland, (iii) the de-synchronisation of Welsh cases after the introduction of annual testing in 2010, and (iv) increasing testing frequency being the desynchronising driver, a result also supported by the outputs of a mechanistic IBM, we can be confident of the interpretation presented here. Our results suggest a distinct shift from high- to low-frequency variability of disease spread after an abrupt interruption in testing.

Population synchrony refers to coincident temporal changes in geographically distinct populations[33]. Desynchronization is desirable for reduction in disease as infectious diseases will eventually die out in small populations unless the pool of infected individuals is refreshed[34]. However, the relationship between synchrony and persistence is highly complex: high dispersal is likely to promote disease persistence, and in addition, high disease incidence (regular outbreaks) may be associated with synchronous spatiotemporal waves of disease. Completely synchronous global fluctuations (Moran effect) can result in extinction. However, local populations extinctions can increase metapopulation persistence[35]. More often than not, there is a negative relationship between the level of synchrony and distance between populations and this has emphasised the importance of dispersal as a synchronising agent[36]. Cattle movement has been shown to spread the disease[4]. Cattle testing, despite being imperfect[32], reduces disease spread as suggested by modelling studies[15,37] and in addition, desynchronises new infections. Vaccinations have been shown to act as desynchronization agents in other diseases[38]. To that end, more research is needed in order to quantify the comparative efficacy of vaccinations and testing in disease spread control and de-synchronisation.

Epidemiological studies in measles showed that repeated (i.e., occurring in a planned and regular manner) and relatively short-lasting events such changes in the crowding of children during school holiday breaks, and/or the associated migration of workers and their families have a significant increase in disease transmission[39,40]. In general, minor changes in the seasonal pattern of disease transmission can lead to massive shifts in the complexity of disease dynamics, possibly resulting in epidemic patterns resembling deterministic chaos[41]. The case examined here is certainly more abrupt than school term breaks, as it lasted considerably longer, the changes in disease control were radical, and the event was unprecedented at that level. To that end, "the emergence of chaotic populations depends subtly on the route to chaos"[41]; even small changes in the epidemiological parameters, reciprocally caused by changes in the epidemic control measures, can lead to chaotic patterns, which are hard to be quantified, predicted, and controlled, as the case examined here. Thus, future control measures or amendments to control measures should be based on designed, gradually-implemented, and causally-predictive interventions.

Do abrupt events synchronise populations? Abrupt or extreme events are generally hard to study as by definition they do not occur frequently. In other biological systems extreme climate events caused synchronised population fluctuations[42]. It has further been suggested that after abrupt events (such as hurricanes), vegetation predictions based on spatial autocorrelation outperform predictions based on temporal autocorrelation[10], indicating strong spatial synchrony. Synchrony after abrupt events has been reported also in climate sciences[8,9]. It therefore merits further investigation to examine to which spatial extend and for how long do such events induce synchronisation and in cases where the synchrony is undesirable, what are the control measures to desynchronize populations.

## Methods

**Data set**. Time series of monthly bovine TB statistics for cattle in GB, publicly available from DEFRA (British Ministry of Food and the Environment) were used. The bTB statistics used here are aggregated at a county scale within GB[28]. Statistics are also aggregated by bTB risk area, which are the high risk, edge, and low-risk area of England as defined by DEFRA. Each county falls into one of these categories based on the epidemiology of the disease in that location[28]. Most counties were characterised as high, edge, or low-risk areas uniformly within them. However, there existed a few counties that were not characterised uniformly regarding the risk status within the area of the county; these counties were further spatially partitioned regarding their risk stratus and appear within the data set regarding their risk status. For example, Derbyshire appears twice in the data set as Derbyshire Edge as well as Derbyshire High. This resulted in a total spatial replicate of 100 single-risk-value counties within GB[28]—thereby counties. The epidemiological risk-based county classification is the one of the data publisher (DEFRA) and refers to the county's risk classification during the year 2016 throughout the data set. Therefore, the risk status is not "dynamic" in the sense that a county classified as high risk in 2016 may have been low risk in another year, but it would still appear as high risk throughout the data. However, such a feature is not feasible with the published data format. A GIS shape file of GB was used to extract the coordinates of the centroid of each risk-based county in UTM format in order to conduct spatial analysis. We analysed the data from January 1996 to end of August 2016 at a monthly time step.

Statistics are affected by seasonal patterns and variations in the frequency of testing; more herds are tested in the winter when more cattle are housed inside. The animals tested are not a random sample of the whole GB herd: herds are tested more frequently in areas of higher bTB incidence than in those with a historically low incidence, and thus herds experiencing several separate incidents in the same reporting period will appear more often in the data set. In general, the more frequent the testing, the more bTB infected herds are likely to be found i.e., purposive sampling is applied[43]. New Herd Incidents (NHI) were used for comparisons between counties in GB[44]. NHI were divided by the number of Total Tests on Herds (TTH) in order to normalise disease spread by testing effort (NHI/TTH). NHI are herds, which were previously officially bTB-free but either had cattle that reacted to a tuberculin test or had a tuberculous animal disclosed by routine meat inspection at slaughter, during the recorded month. TTH is the number of the total tests on each herd on the county on each time step. TTH exclude gamma interferon tests, nonetheless responsible for only a small proportion of new breakdowns. TTH do not account for the contribution of slaughterhouse cases. However, slaughterhouse cases will initiate a herd skin test after being traced back. Details of the methodologies and revisions of the methodologies through time can be found at[43] and references therein. The raw data of NHI, TTH, and NHI / TTH are plotted in Fig. 1a–c, respectively. Seasonality analysis[45] of the NHI/TTH indicator is provided in the Supplementary Note 1. A risk assessment of the NHI/TTH index with bTB incidence is provided in Supplementary Note 7 and Supplementary Fig. 9.

In October 2009, Scotland was officially declared as bTB-free while the rest of GB was not[46]. Scotland has had a risk-based surveillance testing policy[47], under which high-risk herds are tested frequently, while lower-risk herds are tested every 4 years (see also Supplementary Note 2). Wales has had an at least annual testing policy of all herds since January 2010. In England, annual testing is required unless the percentage of infected herds in a state or region of the state is 1% or less. When the percentage of infected herds is 0.2% or <0.1% testing may be conducted every three or 4 years respectively. In practice, most places in the GB tested cattle every 4 years during the 1990s[2], and at least before the introduction of annual testing ~80% of cattle were never tested throughout their life time[18]. The number of herds tested annually has increased since 2010 and expanded further in the East and North English regions in 2011. Low-risk areas in England are areas with percentage of infected herds less than 0.2% and thus tested every 4 years[48,49]. Since 2013, in England herds in the high-risk areas and the edge areas are tested annually, except the edge area part of Cheshire County, where it is 6-monthly.

Throughout the analysis 'before interruption' data include data from January 1996 to December 2000 and 'after interruption' data includes values from November 2002 to August 2016 all in monthly time steps. 'Testing interruption' data include values from January 2001 to (including) October 2002. This is because from January 2001 to November 2001 cattle testing against bTB was significantly reduced due to an outbreak of the FMD in cattle in GB[50]. From November 2001 to October 2002, testing resumed but it was concentrated on clearing the backlog of overdue tests[50]. Therefore, data from January 2001 to October 2002 are distinctly different from other periods[50].

**Empirical probability distributions**. We have plotted the PDF of NHI/TTH for GB (Fig. 1d) as well as high-risk (Fig. 1e) and low-risk (Fig. 1f) areas before and after testing interruption (interruption months are discarded). NHI/TTH is the ratio of the number of outcomes in which a specified event occurs to the total number of trials. The empirical probability density function PDF is then estimated by fitting a kernel function[51].

**Wavelet analysis**. Wavelet analysis is a powerful tool in time series decomposition. A key strength of the wavelet approach is its decomposition properties related to time-scale localisation[52]. Wavelet analysis is notably free from the assumption of stationarity and to that end particularly relevant to the analysis of non-stationary systems, i.e., systems with short-lived transient components, like those observed in ecological systems[52]. Wavelet analysis assumes no stationarity in the data, and it is especially suitable for following gradual change in forcing by exogenous variables[52]. Here we follow the methodology of[53] on "mean-field wavelet", to estimate to examine and quantify the time evolution of spatial synchrony of NHI/TTH in GB. This determined by the number of identical transforms N. We performed mean-field wavelet transform analysis[12,13,54] of NHI/TTH in GB for the whole period (Fig. 2a), GB before testing interruption (Fig. 2b), GB after testing interruption (Fig. 2c), English high-risk areas for the whole data period (Fig. 2d), and English low-risk areas (2e) for the whole data period. The reason for applying the method to two individual periods of the time series is that the interruption interval contributes a large amount of signal power to the analysis and could bias the spectral features of the time series. The effect of the interval is evident in Supplementary Fig. 5. For the wavelet transformation, the Morlet mother wavelet function has been used, the number of oscillations falling within the wavelet envelope width was set to $f_0 = 1$ and no initial de-trending of the time-series was performed. Analysis was conducted in 'reumannplatz' package in R[55].

**Spatial synchrony and network analysis**. We sought to quantify the spatial variation (at the level of county) in patterns of synchrony[14] across time. We calculated correlation matrixes of differences in NHI/TTH per GB counties after the within the county annual variation has been removed[56]. Correlation matrices allow a readily comprehension of variable relationships by reducing high-dimension multivariate structure and are thus the basis for classical multivariate techniques. The data were temporally partitioned into five snapshots, with a criterion of approximately equal number of months per bin, as following: (a) 1/1996–3/2000, (b) 4/2000–4/2004, (c) 5/2004–6/2008, (d) 7/2008–7/2012, and (e) 8/2012–8/2016. Period (b) is a superset of the FMD outbreak and testing interruption months. All correlation matrices figures are plotted in the Supplementary material 'Spatial synchrony and Network analysis'.

We sought to quantify how NHI/TTH evolved spatio-temporally and how the disease spread from high to edge or low-risk areas after testing interruption. Each county's centroid coordinates were used to calculate a non-directed graph between counties[57]. In order to calculate spatial synchrony after the "within the year variation" has been removed within each county, the full data set was used. However, due to the very large number of spatial replicate (counties) and temporal data for each year, only counties with at least six NHI months >0, and counties with (NHI/TTH) > 0.3 are plotted (threshold counties). This plotting threshold is arbitrary, however, results are similar if the plotting threshold is set to at least three NHI months >0, and counties with (NHI/TTH) > 0.4 (results not shown here). Thus, each correlation matrix in (Supplementary Fig. 6a–e) corresponds to each snapshot of the network with the same order as plotted in Fig. 3a–e (i.e., the network in Fig. 3a has a correlation matrix as plotted in Supplementary Fig. 6a). The thickness of the connecting line between two nodes is proportional to the correlation value between those counties in the corresponding correlation matrix. Only positive correlations are plotted. Analysis was conducted in R[55].

The foreground data presented on Fig. 3 were created by the authors using the methods described here. The background map contains Ordnance Survey (OS) data ©Crown copyright and database right (2016) and it is based on the "Ceremonial County boundaries of England", a polygon data set showing each current English County as defined by the Lord Lieutenancies Act 1997. This current version of the data (updated on 10 February 2016) was retrieved on January 2018 from data.gov.uk where it is released under Ordnance Survey Open Data License. More information about the background map can be found here: https://data.gov.uk/dataset/0fb911e4-ca3a-4553–9136-c4fb069546f9/ceremonial-county-boundaries-of-england.

**Individual-based model and comparisons against data**. A calibrated IBM[15] of bTB was used to provide a potential mechanistic reproduction of the dynamics of bTB in cattle regarding the impact of testing frequency by parameterising the IBM for regions of GB and validating model outputs against data regarding testing frequency. The IBM was parameterised for high risk and edge areas in England after testing interruption (November 2002 to August 2016). From January 2013 onwards, annual testing was implemented in these areas. Initially the IBM was parameterised to mimic the annual testing introduction intervention by switching to annual testing in the corresponding to January 2013 simulated month. We sought to quantify the potential deviance between time series data for high risk and edge areas of England and IBM outputs both in terms of NHI/TTH. For this purpose, Bayesian inference time series intervention analysis[58] was employed with data used a control and IBM outputs as the test data set. This served as a null model[16]. Sequentially, IBM testing frequency was set to four years throughout the simulation period (i.e., not switching to annual testing in 2013) and, all else been equal, compared with the same data as before in order to quantify the counterfactual[16,58]. The IBM description, parameterisation, simulation scenarios and Bayesian inference statistical analysis and rationale are described in detail in Supplementary Note 5.

**Data availability statement**. All the data are attached as an excel file in Supplementary Data 1 in the format used in the analysis and are also publicly available at: https://www.gov.uk/government/statistics/incidence-of-tuberculosis-tb-in-cattle-in-great-britain#history.

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

## Acknowledgements

We are thankful to the numerous scientists associated with data collection processing and online repository. Comments from Tim Benton considerably improved earlier versions of the manuscript.

## Author Contributions

A.M. conceived the idea, led the writing, and coordinated research. Y.M., I.N.D., and A. M. analysed the data. Y.M., M.R.E., and I.N.D. helped develop the idea and contributed to the writing of the paper.

## Additional information

**Competing interests:** The authors declare no competing interests.

