## [Peer Review File · Nature Communications]

Reviewers' comments:

Reviewer #3; expertise - Spatial aspects of infectious disease dynamics

I am in little doubt that the statistical methodology and inference set out in this manuscript is well thought out, and of a high standard, especially since the revisions suggested by the first round of reviewers have been implemented. To my mind, the basic findings of the study are to confirm that infection cycles get longer and the cycles are more widely synchronised in areas where the disease is more persistent, and that this synchrony can disappear where controls are more effective and spread mechanisms reduced.

I am, however, a little worried by some of the underlying assumptions and interpretations, which though possibly justified, need to be explicitly covered in the text, and taken into account during the discussion of the results. The following points are not really intended as a criticism of the work but do interfere with the interpretation of the results by those of us who read the work with a less (geo) statistical perspective.

The authors suggest that it was the abruptness of interruptions caused by the FMD outbreak that caused the change in spatio temporal characteristics. I can see that the rapid change in breakdown numbers made it easier to detect these changes – but I wonder whether these contrasting properties are simply a reflection of markedly different disease levels, rather than the causes of increase. I suspect that such differences in synchrony and cycle length may well be found in any disease system where there are adjacent high and low risk areas.

The paper is written as if testing itself has a direct impact on the disease: to quote the abstract “abrupt short-lasting event such as the bTB testing interruption can increase new infections markedly”. This is obviously not the case. Testing reveals infected cattle which are culled, and it is the control measures that have the impact. Thus the interruption in testing during the FMD outbreak meant much reduced control measures, which in conjunction with the hugely increased movement of potentially infected cattle during restocking, led to a marked rise in disease levels post 2001. Whilst a minor irritation, this might reflect a worrying disconnect between the mindset driving the analysis and the real world!

The analyses rely on a disease index which is normalised to remove seasonality, and thus allow analysis of monthly as opposed to annual data. Fair enough. The interpretation of the results is written as if this index is a reliable indicator of disease risk and spread. How does it compare in epidemiological terms with the numbers of breakdowns detected or the incidence or prevalence? Does a high index with few tests/cases have the same significance as a high index with many tests/cases? I would have guessed not, in that a high proportion of a large number of cases seems more likely to pose a threat than a high proportion of fewer cases. Is the relevance of synchrony between an index controlling for sampling effort the same as that between a measure like incidence?

I am slightly at a loss as to why the analyses are done for the high risk, edge and low risk areas throughout the study whole period. The implication is that these different areas are fixed in terms of epidemiological risk. Unless I have misunderstood the text, the three regions are those defined by DEFRA to reflect the disease situation in 2016 and thereafter. They have little relevance to the risk during the earlier years of the study period, when in all probability much of the current high risk area

was in effect 'edge' or 'low risk'. If the authors want to look at the separate dynamics in edge and core areas (i.e. areas where the disease is established and spreading) over time, then surely they should change the edge and high risk distributions over time. Might it be worth discussing the change in spatio temporal characteristics of the high risk area as the disease spreads through it?

Reviewer #4; expertise - Bovine tuberculosis

This manuscript analyses the pattern of bovine Tb incidents in Great Britain, using wavelet analyses and other tools to identify patterns of disease occurrence, both in terms of correlations (between different areas) and cyclical patterns, as well as how they are influenced by short term 'catastrophic' events.

The paper is clearly written and as far as I am able to discern (I am not an expert on wavelet analysis, for example though I am broadly aware of what it is and what it can do) the methods seem both appropriate and the analyses well done. Despite this, I cannot unfortunately recommend this paper for publication in Nature Communications, due to the fact that the level of novelty is not that high. The approaches they are using are new for bovine Tb in Great Britain and could be a useful tool - this I do not doubt.

However, the general points that they make (about sudden events, synchronisation and periodicity) are already made by the series of Grenfell led papers on Measles (which also use wavelets, and which the authors cite).

The specific observations are also already known, e.g. the impact of the FMD epidemic, the annual and four yearly patterns (see the Blake and Donnelly paper that they cite) and the spatial correlations between regions, driven by cattle movements (which would of course increase in strength as the underlying incidence increases). These have all been analysed before. Further, the authors really only go so far as to identify the patterns, and not look more deeply at the underlying causes.

Thus while I can see the place for such a paper in a more specialist journal (and I do believe there is of sufficient interest here for the results to be publishable) it does not make the genuinely general, genuinely novel points I would expect for publication in Nature Communications.

Reviewer #5 ; expertise - Wavelet analyses

The authors calculate an index of bovine TB incidence based on number of new cases normalised by number of tests performed. They report a number of changes in the distribution and dynamics of this index since a disturbance caused by the 2001 foot and mouth outbreak, which was accompanied by a brief suspension of testing and dispersal associated with re-stocking.

Testing rates (TTH) and new herd incidents (NHI) are both increasing steadily over time, and testing protocols are complex and shifting. However the index (NCI/TTB) appears to show a step change associated with the 2001 disturbance. I am not a specialist in bTB and it is unclear to me whether this index is a better measure of underlying disease frequency in the population than NHI, or some other NHI-based estimate that takes into account the non-random nature of the testing.

I believe the statement on line 207 about synchrony promoting disease persistence is an oversimplification. Reference 33 is specifically about the dynamics of a particular kind of network. High dispersal is likely to promote disease persistence, and in addition high disease incidence (regular

outbreaks) may be associated with synchronous spatiotemporal waves of disease. However where synchrony is the result of a driver such as a Moran effect completely synchronous global fluctuations can result in extinction, with no possibility of rescue effects. So the relationship between synchrony and persistence is complex.

Among other results the authors report changes in periodic behaviour based on wavelet-transform analysis, on which I will concentrate my comments below.

I support the decision to present wavelet analyses 'before' and 'after' 2001, since the disturbance is so large that it clearly contributes a great deal of signal power to an analysis of the whole timeseries, and the temporal localisation of the wavelet is not so perfect that it is clear which spectral features may be associated with this effect in the whole-time-series transform. The authors cite a software package but do not explicitly state the type of mother wavelet used in the analysis. In particular, the number of oscillations falling within the wavelet envelope width is the parameter determining the temporal resolution and should be stated in the text. $f_0=1$ is most commonly used. Also, is each time-series linearly detrended separately before transformation?

In accordance with a request by referee 2, the authors state that they have adopted a 'mean field' approach following ref 47. However the 'mean field' approach in 47 is a measure of phase and amplitude synchrony between N wavelet transforms based on a frequency-specific normalisation of the mean wavelet transform by the mean power of all wavelet transforms. The result has an RMS value between 0 and 1, with 1 indicating all N transforms are identical. It is not at all clear that the authors have actually performed this normalisation, and they should state exactly how the values in figure 2 are calculated.

The colour surface plots presented in this paper are not accompanied by colour bars to indicate the values associated with the colours in the colour map, and they should be. The absolute value of the mean field is of interest as a measure of synchrony, and features in the plots represent synchronous oscillations. It appears that these authors' plots are actually plots of the wavelet energy of the mean timeseries, or something like that, rather than being the measure of frequency-specific synchrony that the power-normalised mean field is intended to be. The authors declare that they have used significance tests based on the work of Liu and others, which implies that the spectral features are found to be significant by comparison with red noise or a similar surrogate, but the nature of this null hypothesis should be explicitly stated. Such a univariate surrogate should be used to find power features in a transform of a single signal (such as a mean time-series), and would require substantial adaptation to identify significant synchrony in a power-normalised mean field plot. If the actual method applied is simply to transform the mean or total index, and look for spectral features in that, then state it explicitly.

The authors find some kind of feature with a periodicity of 4 years, which appears to be robust to removal of the 2001 disturbance. They also find a feature with a periodicity of 1 year, which need to be reconciled with their working indicating that the index is not seasonally biased.

Overall the presentation of the wavelet results is somewhat vague and confusing, and explicit methodological detail is required, plus an actual scale of colour values on the plots.

Dear reviewers,

we wish to thank you all for your time and effort in reviewing this work. For your convenience all of your text has been marked with *blue colour and italics*, while our response is in black colour normal letter style when it refers to the reviews and in *black colour italicized* when it refers to a new/amended text version in the re-revised manuscript (ms). Throughout the main ms and supplementary material any changes made were marked with track changes for your convenience.

Best regards,

Aris

Reviewers' comments:

Reviewer #3; expertise - Spatial aspects of infectious disease dynamics

I am in little doubt that the statistical methodology and inference set out in this manuscript is well thought out, and of a high standard, especially since the revisions suggested by the first round of reviewers have been implemented. To my mind, the basic findings of the study are to confirm that infection cycles get longer and the cycles are more widely synchronised in areas where the disease is more persistent, and that this synchrony can disappear where controls are more effective and spread mechanisms reduced.

We wish to thank the reviewer for the positive and very thorough reading of our ms. We have incorporated all comments to the best we could as we think that they improve the ms. Regarding the last comment (high-low risk areas) we simply rested by acknowledging and discussing it as unfortunately it is an issue related with the data publisher/curator and there is little we could do about it.

I am, however, a little worried by some of the underlying assumptions and interpretations, which though possibly justified, need to be explicitly covered in the text, and taken into account during the discussion of the results. The following points are not really intended as a criticism of the work but do interfere with the interpretation of the results by those of us who read the work with a less (geo) statistical perspective.

Indeed, we acknowledge that some issues of 'statistical autism' may have been present in our ms. Following the suggestions provided by the reviewer we have tried to account for this issue by amending our phrasing.

The authors suggest that it was the abruptness of interruptions caused by the FMD outbreak that caused the change in spatio temporal characteristics. I can see that the rapid change in breakdown numbers made it easier to detect these changes – but I wonder whether these contrasting properties are simply a reflection of markedly different disease levels, rather than the causes of increase. I suspect that such

differences in synchrony and cycle length may well be found in any disease system where there are adjacent high and low risk areas.

Indeed, one could see such differences in synchrony and cycle length in other disease systems where there are adjacent high and low risk areas. However, prior to testing interruption there were very few high risk areas. Then, it is also expected (according to the literature) that even repeated (i.e. occurring planned and regularly) and relatively short lasting events such as the congregation of children during school holiday breaks, and/or the associated migration of workers and their families have a significant increase in disease transmission^{41,42}. In general, minor changes in the seasonal pattern of disease transmission can lead to massive shifts in the complexity of disease dynamics, possibly resulting in epidemic patterns resembling deterministic chaos⁴³. The case examined here is certainly more abrupt than school term breaks as it lasted considerably longer, the changes in disease control were radical, and the event was unprecedented at that level. Even small changes in the epidemiological parameters, reciprocally caused by changes in the epidemic control measures, can lead to chaotic patterns which are hard to be quantified, predicted, and controlled, as the case examined here.

In addition, abrupt events have been reported to be a causal agent of population synchrony: in other biological systems extreme climate events caused synchronized population fluctuations⁴⁴. It has further been suggested that after abrupt events (such as hurricanes) vegetation predictions based on spatial autocorrelation outperform predictions based on temporal autocorrelation¹⁰ indicating spatial synchrony.

Therefore, it is highly unlikely that the patterns recorded here are due to markedly different disease levels. We found this comment to be useful and to that end we added two extra paragraphs at the end of the discussion addressing this.

References:

- Metcalf, C.J.E., Bjørnstad, O.N., Grenfell, B.T. and Andreasen, V. 2009. Seasonality and comparative dynamics of six childhood infections in pre-vaccination Copenhagen. *Proceedings of the Royal Society B: Biological Sciences*, **276**: 4111-4118.
- Dalziel, B.D., Bjørnstad, O.N., van Panhuis, W.G., Burke, D.S., Metcalf, C.J.E. and Grenfell, B.T. 2016. Persistent Chaos of Measles Epidemics in the Pre-vaccination United States Caused by a Small Change in Seasonal Transmission Patterns. *PLoS Comput. Biol.*, **12**: e1004655.
- Grassly, N.C. and Fraser, C. 2006. Seasonal infectious disease epidemiology. *Proceedings of the Royal Society B: Biological Sciences*, **273**: 2541-2550.

The paper is written as if testing itself has a direct impact on the disease: to quote the abstract “abrupt short-lasting event such as the bTB testing interruption can increase new infections markedly”. This is obviously not the case. Testing reveals infected cattle which are culled, and it is the control measures that have the impact. Thus the interruption in testing during the FMD outbreak meant much reduced control measures, which in conjunction with the hugely increased movement of potentially infected cattle during restocking, led to a marked rise in disease levels post 2001. Whilst a minor irritation, this might reflect a worrying disconnect between the mindset driving the analysis and the real world!

The reviewer is of course right here; it is not testing interruption per se that caused the observed pattern, it is the change in epidemiological parameters provided due to the lack of control or major

amendments in the control measures during the FMD outbreak. We have now rephrased both the abstract as well as several other sections of the ms including the discussion specifying that

“It should be noted that it is not testing interruption per se that increases infections; the interruption simply provided (a) time availability for unmanaged disease spread within each farm and between neighbouring farms (fine scale), and (b) time availability for unmanaged disease spread via cattle movements to distant farms (coarser scales). Thus, the lack of control during the bTB testing interruption changed the epidemiological parameters such as infection rate, dispersal distances, and population densities”

The analyses rely on a disease index which is normalised to remove seasonality, and thus allow analysis of monthly as opposed to annual data. Fair enough. The interpretation of the results is written as if this index is a reliable indicator of disease risk and spread. How does it compare in epidemiological terms with the numbers of breakdowns detected or the incidence or prevalence? Does a high index with few tests/cases have the same significance as a high index with many tests/cases? I would have guessed not, in that a high proportion of a large number of cases seems more likely to pose a threat than a high proportion of fewer cases. Is the relevance of synchrony between an index controlling for sampling effort the same as that between a measure like incidence?

We think that reviewer has a good point and we agree that some risk assessment or link to the index employed here (NHI/TTH) with incidence would certainly improve the ms. Given that with the current data format it is not feasible to calculate incidence (neither at the county scale, nor at the level of individuals across the country), we compared NHI/TTH with IBM-derived incidence. We have now added a full new section in the supplement (S7. Quantifying the potential deviance between disease Incidence and NHI/TTH) addressing this.

The analysis indicated that NHI/TTH is at least one level of magnitude larger than incidence. Overall there is a good fit between NHI/TTH and incidence, however there are cases where NHI/TTH, an index with few tests/cases, does not predict incidence, an index with many tests/cases. In order to account for this in the synchrony analysis (Figure 3 in the main text and Figures S4 and S6 in the supplement) only above threshold counties were plotted (counties with at least six NHI months > 0 , and counties with $(\text{NHI/TTH}) > 0.3$).

We finally acknowledge that in general, a high proportion of a large number of cases (incidence) is more likely to pose a threat than a high proportion of fewer cases (NHI/TTH). To that end the results derived throughout are likely to underestimate disease spread and synchrony in comparison to the results that would have been derived had this work used incidence.

I am slightly at a loss as to why the analyses are done for the high risk, edge and low risk areas throughout the study whole period. The implication is that these different areas are fixed in terms of epidemiological risk. Unless I have misunderstood the text, the three regions are those defined by DEFRA to reflect the disease situation in 2016 and thereafter. They have little relevance to the risk during the earlier years of the study period, when in all probability much of the current high risk area was in effect ‘edge’ or ‘low risk’. If the authors want to look at the separate dynamics in edge and core areas (i.e. areas where the disease is established and spreading) over time, then surely they should change the edge and

high risk distributions over time. Might it be worth discussing the change in spatio temporal characteristics of the high risk area as the disease spreads through it?

We clearly see the rationale of the reviewer here and at earlier stages of our analysis we have attempted to use dynamically-defined risk status changing every year. However, we realized that this is not feasible with this data published format. In detail:

The data are published and curated by DEFRA and they are published in this format: every county appears in the dataset at the level of epidemiological risk. In practice this means two things: (i) if a fraction of the county is high risk and a fraction edge, or low risk end edge, then this county will appear twice in the dataset once as e.g. 'Derbyshire Edge' and once as 'Derbyshire High' with the corresponding epidemiological values for that fraction of the county. Alternatively, if all the county is under a single epidemiological risk status (all of it high, edge, or low) then it will appear once in the dataset. Indeed, this classification is the one of 2016. So the first point is that we have to use this epidemiological status (high, edge, low) to distinguish between the same county appearing twice in the data under different epidemiological status, as this is the way the data are published.

We had attempted to do exactly that early on in our analysis however this is simply not feasible with the current dataset format; indeed, the risk assessment format is the one published by DEFRA for the year 2016. Indeed, the one of year 2013 or indeed any other year was slightly or very different etc. However, the catch here is that if one wants to use 'dynamic' risk status areas changing every year, one would also need to change the spatial replicate of the dataset: for example, in 2016 the county Derbyshire in 2016 appears as 'Derbyshire Edge' and as 'Derbyshire High'. In e.g. 2008 it may have been all edge and therefore it would appear as 'Derbyshire Edge' all of it and Derbyshire High not have been in the dataset. That means that our spatial sample size in 2008 would have been smaller than the one in 2016. The statistics that we use could not cope with different sample sizes per time step. In addition to that, a county that appears in the dataset under only one risk category (as it happens with the vast majority of counties) may be in 'high risk' status in 2016 while it may have been in 'low risk' status in 1997. However, the way the data are published we wouldn't be able to distinguish if a county characterised as high risk in 2016 would have been characterised as such or how would it have been characterised in earlier years.

Therefore, while we clearly see the reviewer's rationale here there is very little we can do about it but state it clearer in the data description section of the methods in the main document.

=====

Reviewer #4; expertise - Bovine tuberculosis

This manuscript analyses the pattern of bovine Tb incidents in Great Britain, using wavelet analyses and other tools to identify patterns of disease occurrence, both in terms of correlations (between different areas) and cyclical patterns, as well as how they are influenced by short term 'catastrophic' events.

The paper is clearly written and as far as I am able to discern (I am not an expert on wavelet analysis, for example though I am broadly aware of what it is and what it can do) the methods seem both appropriate and the analyses well done. Despite this, I cannot unfortunately recommend this paper for publication in Nature Communications, due to the fact that the level of novelty is not that high. The approaches they are using are new for bovine Tb in Great Britain and could be a useful tool - this I do not doubt.

We wish to thank the reviewer for the critical review evaluation. Below we show why we think that the reviewer's statement regarding novelty is not valid. However, the review provided was useful for us, as it motivated us to re-read all the mentioned papers which helped us improve the discussion and the novelty of our work in a more comprehensive way.

However, the general points that they make (about sudden events, synchronisation and periodicity) are already made by the series of Grenfell led papers on Measles (which also use wavelets, and which the authors cite).

These papers are part of the canon in this field and they were cited as the reviewer states. After re-reading these papers we were still unable to find reference to 'sudden' or 'abrupt' or 'extreme' or 'rare' events in those papers and therefore the reviewer's statement that: *'the general points that they make (about sudden events, synchronisation and periodicity) are already made by the series of Grenfell led papers on Measles'* seems to us to not be valid. The point behind our study here is the impact of an abrupt event on population synchrony.

In addition, in those papers (which are about Measles as the reviewer also states) the control measure is vaccination - not testing and culling as is the case in our study. The efficacy of vaccination vs. testing and culling is a very timely topic and to that end it is not trivial (we should point out that we would not advocate testing and culling as a control measure for childhood measles). For this additional reason we feel that the reviewer's statement regarding novelty is also not valid.

References:

- Viboud, C., Bjørnstad, O.N., Smith, D.L., Simonsen, L., Miller, M.A. and Grenfell, B.T. 2006. Synchrony, waves, and spatial hierarchies in the spread of influenza. *Science*, **312**: 447-451.
- Grenfell, B., Bjørnstad, O. and Kappey, J. 2001. Travelling waves and spatial hierarchies in measles epidemics. *Nature*, **414**: 716-723.
- Bjørnstad, O.N. and Grenfell, B.T. 2001. Noisy clockwork: Time series analysis of population fluctuations in animals. *Science*, **293**: 638-643.

The specific observations are also already known, e.g. the impact of the FMD epidemic, the annual and four yearly patterns (see the Blake and Donnelly paper that they cite) and the spatial correlations between regions, driven by cattle movements (which would of course increase in strength as the underlying incidence increases). These have all been analyses before. Further, the authors really only go

so far as to identify the patterns, and not look more deeply at the underlying causes. Thus while I can see the place for such a paper in a more specialist journal (and I do believe there is of sufficient interest here for the results to be publishable) it does not make the genuinely general, genuinely novel points I would expect for publication in Nature Communications.

The Blake & Donnelly paper concerns the analysis of the data on TB cases, they critique the current DEFRA method and suggest an alternative. Their motivation appears to be to allow a determination of whether control measures are actually having any effect. The observed data as presented and the fitted model in Blake & Donnelly do appear to show some cycles but is not a big part of the paper, in fact we cannot find any discussion or analysis of the cyclical pattern - the paper discusses the ratio of high risk herds assigned to annual testing and the probability of previously TB-free herds to have their TB-free status removed after a 4-year test, which are different altogether from what we do. This is a methods paper designed to demonstrate a better method of assessing infection risk, not to quantify temporal cycles. The particular difference (and novelty) between our ms and Blake & Donnelly, is that their paper does not detect changes before and after testing in fact it does not contain data from before the testing interruption (the span of their data is 2003-2011, while the one analysed here is 1996-2016, testing interruption was in 2001). Therefore, one of the key issue in our ms - showing that the 1-year cycle prior to interruption shifted to 4-year cycles after interruption is simply not possible in the Blake & Donnelly paper. To our knowledge it has not been shown that infection cycles are becoming longer after an abrupt event. Therefore, in our view the reviewer's statement '*The specific observations are also already known, e.g. the impact of the FMD epidemic, the annual and four yearly patterns (see the Blake and Donnelly paper that they cite)*' is not reasonable.

Regarding spatial aspects of cattle movements, there exist several papers indicating that cattle movements are a key agent of bTB disease spread, but to our knowledge there is no paper quantifying '*the spatial correlations between regions*' as the reviewer states. There are indeed several papers quantifying either cattle movements alone or cattle movements and bTB disease spread. However, to our knowledge none of these studies 'spatial correlation' between locations and most importantly (which is key to this study) 'population' synchrony' – all these elements are simply not there. By doing so here, this study goes well beyond '*identifying the patterns*' as the reviewer states. Please note that quantifying cattle movement or spatial disease incidence or prevalence on a map is a different thing altogether from spatial cross-correlation, or population synchrony.

Lastly, the reviewer states that '*and not look more deeply at the underlying causes*'. As already evident from the title this study aims to identify the impact of an abrupt event on spatio-temporal disease spread and population synchrony. While we do acknowledge that identifying the causes of the disease is an important question, we never claimed to do so, the reviewer is really asking us to conduct another study altogether. However, the study that we did conduct, the impact of an abrupt event on space-time disease spread, is also a very important question and this study aimed to contribute to this question. Our study goes well beyond identifying patterns; we have applied state-of-the-art individual based modelling (IBM) together with the latest causal impact Bayesian statistical inference assessing the causality of our findings. The statement that '*this study did not look more deeply at the underlying patterns*' is really not valid.

On the positive side, following the reviewer's comments we strived to improve the presentation of the novelty of the findings in the ms and present them in a more comprehensive manner. Additional reading of other papers of measles, made us realize that even a short lasting and fairly minor changes in the epidemiological parameters (in comparison to the one examined here which is abrupt), such as school holiday breaks, was shown to increase infections. The event examined here is unique in comparison to school term breaks and we highlight this now better in the ms. However, we wish to thank the reviewer for the criticism that motivated us to read further and improve our discussion as well as present the findings in a more comprehensive manner.

References to school term holidays:

- Metcalf, C.J.E., Bjørnstad, O.N., Grenfell, B.T. and Andreasen, V. 2009. Seasonality and comparative dynamics of six childhood infections in pre-vaccination Copenhagen. *Proceedings of the Royal Society B: Biological Sciences*, **276**: 4111-4118.
- Dalziel, B.D., Bjørnstad, O.N., van Panhuis, W.G., Burke, D.S., Metcalf, C.J.E. and Grenfell, B.T. 2016. Persistent Chaos of Measles Epidemics in the Prevaccination United States Caused by a Small Change in Seasonal Transmission Patterns. *PLoS Comput. Biol.*, **12**: e1004655.
- Grassly, N.C. and Fraser, C. 2006. Seasonal infectious disease epidemiology. *Proceedings of the Royal Society B: Biological Sciences*, **273**: 2541-2550.

=====

Reviewer #5 ; expertise - Wavelet analyses

The authors calculate an index of bovine TB incidence based on number of new cases normalised by number of tests performed. They report a number of changes in the distribution and dynamics of this index since a disturbance caused by the 2001 foot and mouth outbreak, which was accompanied by a brief suspension of testing and dispersal associated with re-stocking.

We wish to thank the reviewer for the positive, technically sound and thorough evaluation of our ms. We are particularly thankful for detecting points that were not clear enough in the previous ms version regarding wavelets. We have implemented all comments as in our view they all improve the ms.

Testing rates (TTH) and new herd incidents (NHI) are both increasing steadily over time, and testing protocols are complex and shifting. However the index (NCI/TTB) appears to show a step change associated with the 2001 disturbance. I am not a specialist in bTB and it is unclear to me whether this index is a better measure of underlying disease frequency in the population than NHI, or some other NHI-based estimate that takes into account the non-random nature of the testing.

Please see also (a) section 'S1 Data – Seasonality of NHI/TTH', in the supplementary material, (b) our reply to the (previous) reviewer #1, as well as (c) our reference to purposive sampling and the statistical sampling theory behind it in the main text section: 'Methods, Dataset', second paragraph starting with: 'Statistics are affected by seasonal patterns...'. In addition to that following a comment of rev#3 (the relationship between incidence and NHI/TTH, we have added a new section Supplementary material S7 providing some risk assessment between the index employed here and incidence.

I believe the statement on line 207 about synchrony promoting disease persistence is an oversimplification. Reference 33 is specifically about the dynamics of a particular kind of network. High dispersal is likely to promote disease persistence, and in addition high disease incidence (regular outbreaks) may be associated with synchronous spatiotemporal waves of disease. However where synchrony is the result of a driver such as a Moran effect completely synchronous global fluctuations can result in extinction, with no possibility of rescue effects. So the relationship between synchrony and persistence is complex.

We are grateful to the reviewer for pointing this out and in fact we found it very well phrased, so we introduced part of the reviewer's comment into the main text acknowledging the complexity of this phenomenon.

Among other results the authors report changes in periodic behaviour based on wavelet-transform analysis, on which I will concentrate my comments below.

I support the decision to present wavelet analyses 'before' and 'after' 2001, since the disturbance is so large that it clearly contributes a great deal of signal power to an analysis of the whole timeseries, and the temporal localisation of the wavelet is not so perfect that it is clear which spectral features may be associated with this effect in the whole-time-series transform.

We wish to thank the reviewer for this. As stated in our reply to the previous rev1 and rev2, the reason for applying the method to two individual periods of the time series, is that the interruption interval

contributes a large amount of signal power to the analysis and could bias the spectral features of the time series. We have also highlighted this comment in the methods, wavelets section as it may not be clear to a non-expert reader.

We have also incorporated this comment in the Methods section in the revised manuscript

The authors cite a software package but do not explicitly state the type of mother wavelet used in the analysis. In particular, the number of oscillations falling within the wavelet envelope width is the parameter determining the temporal resolution and should be stated in the text. $f_0=1$ is most commonly used. Also, is each time-series linearly detrended separately before transformation?

We acknowledge that our description was not detailed enough and we expanded on it as the reviewer suggests. The Morlet mother wavelet function has been used with $f_0=1$. Dog and Paul mother wavelets have also been used with similar results (not shown here). The time-series were not linearly de-trended before the transformation. This information has been included in the revised manuscript.

In accordance with a request by referee 2, the authors state that they have adopted a 'mean field' approach following ref 47. However the 'mean field' approach in 47 is a measure of phase and amplitude synchrony between N wavelet transforms based on a frequency-specific normalisation of the mean wavelet transform by the mean power of all wavelet transforms. The result has an RMS value between 0 and 1, with 1 indicating all N transforms are identical. It is not at all clear that the authors have actually performed this normalisation, and they should state exactly how the values in figure 2 are calculated.

Our adaption of the method of ref 47 involves the estimation of the aggregated time series for each high, low risk and total GB. We followed this approach due to the large number of spatial replicate (counties) and temporal data for each year with NHI/TTH equal to zero. In the revised manuscript, we have added the explicit methodology suggested by ref 47, which as the reviewer #5 suggests, also offers a quantification on the number of identical transforms N. This analysis confirms the periodicities estimated by wavelet analysis of the means, as the patterns present very strong similarities.

The colour surface plots presented in this paper are not accompanied by colour bars to indicate the values associated with the colours in the colour map, and they should be. The absolute value of the mean field is of interest as a measure of synchrony, and features in the plots represent synchronous oscillations. It appears that these authors' plots are actually plots of the wavelet energy of the mean timeseries, or something like that, rather than being the measure of frequency-specific synchrony that the power-normalised mean field is intended to be.

Indeed, the plots in the original manuscript depicted the wavelet energy of the mean time series. In the revised manuscript, we have added colour bars to associate them with the map colours. In the revised manuscript, the mean field approach was also included (in the supplementary material section S3, Fig. S3c), where the colours present the fraction of identical N wavelet transforms (please see also previous comment).

The authors declare that they have used significance tests based on the work of Liu and others, which implies that the spectral features are found to be significant by comparison with red noise or a similar

surrogate, but the nature of this null hypothesis should be explicitly stated. Such a univariate surrogate should be used to find power features in a transform of a single signal (such a mean time-series), and would require substantial adaptation to identify significant synchrony in a power-normalised mean field plot. If the actual method applied is simply to transform the mean or total index, and look for spectral features in that, then state it explicitly.

The null hypothesis of an auto-regressive lag-1 model was used with autocorrelation coefficient of lag-1 $\rho_0 = 0.3$. This was estimated from the autocorrelation function of the empirical data for all the wavelet transforms except the Low Risk regions, where the white noise model was used ($\rho_0 = 0$).

The authors find some kind of feature with a periodicity of 4 years, which appears to be robust to removal of the 2001 disturbance. They also find a feature with a periodicity of 1 year, which need to be reconciled with their working indicating that the index is not seasonally biased.

Thank you for pointing this out, as it certainly needs to be further clarified in the manuscript. The fact that the index used is the ratio of New Herd Incidents vs the number Total Tested Herds, compensates for the seasonality in testing, which is substantially biased in winter when the cattle are kept indoors. However, other factors affecting annual seasonality might arise, e.g., the effect of climatic factors in the spread of the bTB. Hence, in the revised manuscript we clarify that the index is not biased by the testing seasonality.

Overall the presentation of the wavelet results is somewhat vague and confusing, and explicit methodological detail is required, plus an actual scale of colour values on the plots.

Following the reviewer's constructive suggestions, the results session has been reformulated to:

Mean Field Wavelet analysis of NHI/TTH suggests at least two significant cycles with strong spatial coherency (Fig. 2) and the frequent emergence of high-frequency synchronization events (95% confidence level across all three significance wavelet tests^{1,2}). The event of testing interruption manifests as the strongest signal when the whole record is taken into account (Fig. 2a). Before testing interruption the annual fluctuation of the infection cycle was dominant (Figure 2b), which gradually, after the abrupt event, moved to lower frequencies, dispersed around a 4-year mean (Fig. 2c). The high-frequency synchronization can be found in the 2- to 4-month band and its duration ranges from couple of months to half a year. For high risk areas, the synchronicity in some events reaches 1, i.e., there is complete synchronization of all the counties (Fig. 2d). The weaker 12-month cycle also persisted, but only for one third of the low risk areas, where both the high and the low frequency co-variability is negligible (Fig. 2e). It is noteworthy that after 2012 the high-frequency synchronization effects decline abruptly to low levels for both high and low risk areas.

We hope that the presentation of the wavelet analysis and results are now more explicit and clear.

REVIEWERS' COMMENTS:

Reviewer #3 (Remarks to the Author):

As for the first round, I am simply not qualified to judge the accuracy/appropriateness of the detailed statistics, and I suspect that the decision to publish should largely rest on such criteria.

As far as I can see the authors have addressed my earlier concerns - and reduced the level of "statistical autism" (a term I shall adopt myself!).

I am still somewhat concerned about the static analysis zones (edge, high and low risk) - which are really applicable to the situation prior to about 2002 or 2003 because the disease has spread so much since then, but I guess we have to accept that the analyses are not amenable to dynamic zonation. A pity, but not enough to reject publication unless the statistician reviewers think otherwise

Reviewer #5 (Remarks to the Author):

The wavelet analysis portion of the text is now substantially clearer. The mean field analysis shows synchronous features associated with significant fluctuations in the mean time series (this is now shown in supp. mat).

To eliminate any ambiguity I suggest the following revision to line 96 of the revised document (line 98 of pdf with track changes):

Wavelet mean field analysis of NHI/TTH (Fig. 2) suggests the frequent emergence of high frequency synchronization events and synchronous features associated with two slow, statistically significant fluctuations in the mean of all time-series (Fig S3iii in Supplementary Materials).

I have no further comments on the wavelet analysis.